# SARS-CoV-2 Transmission in Belgian French-Speaking Primary Schools: An Epidemiological Pilot Study

**DOI:** 10.3390/v14102199

**Published:** 2022-10-06

**Authors:** Julie Frère, Olga Chatzis, Kelly Cremer, Joanna Merckx, Mathilde De Keukeleire, Florence Renard, Nathalie Ribesse, Frédéric Minner, Jean Ruelle, Benoit Kabamba, Hector Rodriguez-Villalobos, Bertrand Bearzatto, Marie-Luce Delforge, Coralie Henin, Fabrice Bureau, Laurent Gillet, Annie Robert, Dimitri Van der Linden

**Affiliations:** 1Pediatric Infectious Diseases, Pediatric Department, CHU Liège, 4000 Liège, Belgium; 2Pediatric Infectious Diseases, Specialized Pediatric Service, Pediatric Department, Cliniques Universitaires Saint-Luc, 1200 Brussels, Belgium; 3Department of Epidemiology and Biostatistics, Institut de Recherche Expérimentale et Clinique, Faculty of Public Health, UCLouvain, 1200 Brussels, Belgium; 4Department of Epidemiology, Biostatistics and Occupational Health, McGill University, Montreal, QC H3A 1A2, Canada; 5Office de la Naissance et de l’Enfance (ONE), 1060 Brussels, Belgium; 6Immunology-Vaccinology Lab of the Faculty of Veterinary Medicine, ULiège, 4000 Liège, Belgium; 7Pôle de Microbiologie Médicale (MBLG), UCLouvain, 1200 Brussels, Belgium; 8SmartGene Services, EPFL Innovation Park, 1015 Lausanne, Switzerland; 9Department of Microbiology, Cliniques Universitaires Saint-Luc, 1200 Brussels, Belgium; 10Institut de Recherche Expérimentale et Clinique (IREC), Center for Applied Molecular Technologies (CTMA), UCLouvain, 1200 Brussels, Belgium; 11Institut de Biologie Clinique de l’Université Libre de Bruxelles, 1070 Brussels, Belgium; 12Federal Testing Platform for COVID-19, Université Libre de Bruxelles, 1070 Brussels, Belgium

**Keywords:** SARS-CoV-2, schools, children, COVID-19, saliva testing, transmission

## Abstract

Schools have been a point of attention during the pandemic, and their closure one of the mitigating measures taken. A better understanding of the dynamics of the transmission of SARS-CoV-2 in elementary education is essential to advise decisionmakers. We conducted an uncontrolled non-interventional prospective study in Belgian French-speaking schools to describe the role of attending asymptomatic children and school staff in the spread of COVID-19 and to estimate the transmission to others. Each participant from selected schools was tested for SARS-CoV-2 using a polymerase chain reaction (PCR) analysis on saliva sample, on a weekly basis, during six consecutive visits. In accordance with recommendations in force at the time, symptomatic individuals were excluded from school, but per the study protocol, being that participants were blinded to PCR results, asymptomatic participants were maintained at school. Among 11 selected schools, 932 pupils and 242 school staff were included between January and May 2021. Overall, 6449 saliva samples were collected, of which 44 came back positive. Most positive samples came from isolated cases. We observed that asymptomatic positive children remaining at school did not lead to increasing numbers of cases or clusters. However, we conducted our study during a period of low prevalence in Belgium. It would be interesting to conduct the same analysis during a high prevalence period.

## 1. Introduction

Schools have been the subject of many controversies during the COVID-19 pandemic. Local infection prevention and control (IPC) measures differed between countries. While some countries chose for a zero-COVID strategy from the beginning [1], others (such as many European countries) took public health measures aiming at controlled circulation of the virus. Those countries had to find the right balance between the competing risks for society of hospitals, and especially intensive care units, under pressure; the risks related to the lockdown of several sectors (e.g., culture); and that of a more genuine focus on the wellbeing of children by keeping schools open [2]. 

In some parts of the world, on the contrary, school closure was chosen as a major strategy to slow down the epidemic. Most reviews show, however, the major downside of this mitigation measure [3], because it dramatically impacts education and both the physical and mental health of millions of children. Many therefore concluded that schools should be the last institutions to close [4,5].

Schools are complex, open, and interconnected systems with the outside world and its communities. By consequence, the dynamic of transmission of SARS-CoV-2 within schools is highly dependent on several factors, such as: virus incidence in the community [6], circulating virus variants (i.e., variants of concern), household transmission [7,8], transport use [9], population density [10], traveling, environmental factors (e.g., temperature), participation in extracurricular activities, social determinants of participants (e.g., ethnicity) [11] and the application of mitigations measures such as mask use (by teachers and/or pupils) [12], ventilation [13] and vaccination coverage [14]. 

Both our knowledge and the dynamics themselves of SARS-CoV-2 transmission between children and from children to adults have evolved during the crisis, especially with the emergence of new variants. Although children were initially considered as super-spreaders in similitude to influenza epidemics [15], accumulating data suggest that they did not play a major role early in the pandemic [16]. However, a study performed in a primary school during the second wave in Belgium shows a similar pattern of transmission (i) between children and (ii) between children and teachers or employees, suggesting that transmission from children might have been higher at later stages than suspected [17]. 

The general objective of this study is to describe the dynamics of transmission of SARS-CoV-2 in elementary education in the French community (named Fédération Wallonie-Bruxelles (FWB)), Belgium, during winter and spring 2021. 

## 2. Materials and Methods

An uncontrolled non-interventional prospective study, named DYNAtracs, DYNAmic of TRAnsmission of Coronavirus in Schools, was conducted between January 2021 and May 2021, in Belgian French-speaking elementary schools (children between 6 and 12 years old) from the FWB. This sub-study is part of a larger study exploring the dynamics of transmission of SARS-CoV-2 in elementary schools and the wellbeing of children during the pandemic. The specific objectives of this sub-study were to describe, during the second (i.e., starting on week 36, 2020) and third (week 7 to week 25, 2021) waves (Figure 1) [18] of the pandemic when the Alpha variant dominated, the role of asymptomatic children attending primary schools in the spread of SARS-CoV-2, and to estimate the incidence of children who were infected through contact with an index case outside of the school environment. In addition, the study was designed to assess the acceptability of a salivary sample taken with a swish/gargle technique in children. 

### 2.1. Study Setting and Participants Enrolment

Among Belgian elementary schools from the FWB, we identified a representative sample using purposive sampling according to three surrogate markers: (i) school size, (ii) socio-economic status (SES) of the pupils attending the schools, and (iii) cumulative incidence of SARS-CoV-2 infection in the geographic province of the school during the first wave of the pandemic (spring 2020). 

More specifically, we applied the FWB’s definition of a small versus large school, i.e., a school with a lower hosting capacity, or of more than 230 pupils, in order to consider the contacts within the school. A value for SES, measured using the FWB’s official 20-point-scale index for all schools, equal or greater than 13 (upper tertile) was defined as high, and a value lower or equal to 7 (first tertile) was defined as low. A low versus high incidence region was described as a province with a cumulative municipal incidence of less than 5.0/1000 persons versus more than 5.0/1000 persons, on 6th May 2020. Investigators refer to the publicly available Belgian epidemiological reports from Sciensano, the Belgian institute of health [19]. Crossing the three criteria resulted in the definition of eight categories of schools, from which the sample was selected. Schools were included sequentially, and upon acceptance, if an invited school declined, another school meeting the same criteria was invited to participate. 

All children registered and attending the participating schools, along with all adults working inside the schools, were then invited to participate through a letter. For each school, investigators provided information about the nature of the study to the children’s legal representatives—parents or guardians—and to school staff, during an online live meeting. A website with details about the study design, study information, informed consent, and videos on how to perform a mouth rinse/gargle specimen was available for all (potential) participants (https://www.sesa.ucl.ac.be/Dynatracs/, accessed on 25 August 2022). The informative letters were translated from French into the most encountered languages in the school communities. 

Only children who provided informed consent signed by their legal guardian and staff who signed the consent were included in the study. 

### 2.2. Study Design, Sample Collection and Laboratory Analysis

Pupils and staff included in the study population were followed-up during six weekly visits over subsequent weeks, according to the school calendar. Therefore, school breaks were not included in the study time. The start date differed between schools due to differences in timeline for obtaining consents, school location, and field planning. On the first visit corresponding to the inclusion visit, a blood sample was taken from all participants by finger prick to perform a rapid serological test (Avioq^®^, Bio-Tech, Shandong China). The CE-labeled Avioq^®^ test is a lateral-flow antibody IgG/IgM test (colloidal gold) that targets the SARS-CoV-2 N-protein. The combined sensitivity for IgM and IgG was 68.8% (CI 95% 60.3–76%) with a specificity of 95.8% (CI 95% 88.5–98.6%) [20]. After collection, serological tests were first read on site by one of three designated study staff and sequentially sent for secondary reading to one single laboratory (Department of Microbiology, Cliniques universitaires Saint-Luc, Brussels) by one experienced microbiologist. During that same first visit, a mouth rinse/gargle specimen for SARS-CoV-2 detection by quantitative real-time reverse transcriptase PCR (RT-qPCR) was also collected. On subsequent weekly visits, all participants provided a mouth rinse/gargle specimen for a total duration of six weeks. Adult participants were invited to self-collect their mouth rinse/gargle specimens while pupils’ specimens were collected under supervision of study staff. All samples were collected on site and in a well-ventilated room or outside depending on the facilities and the meteorological conditions. The supervising study staff was wearing personal protective equipment (PPE) including a FFP2 mask. For each collection, a 5 mL vial of sterile 0.9% saline was squeezed into the participant’s mouth. They were then asked to swish the content for 5 s followed by tilting their heads back and gargling for 5 s. This swish/gargle cycle was repeated two more times and then the saline was expelled into a dedicated device designed by the University of Liège, commercialized by Diagenode (4100 Seraing, Belgium). The sampling kit was equipped with a dosing funnel that permitted the collection of exactly 1.2 mL of saliva, which was subsequently mixed with 2 mL of lysis buffer, inactivating the virus [21]. The self-collected swish/gargle sampling technique was previously evaluated in adults and school-aged children and compared to nasopharyngeal swabbing. It had a better acceptability with a good sensitivity of 97.5% (95% CI 86.8–99.9%) [22]. 

Participants were asked to not drink or eat within the 1 h preceding sampling. In case of sampling failure, a second attempt was not allowed. Saliva samples were directly dispatched to one of the three participating laboratories, attached to the three French-speaking universities, i.e., Department of Microbiology of Cliniques universitaires Saint-Luc, Federal testing platform COVID-19 of the Université Libre de Bruxelles, and the COVID-19 laboratory of the University of Liège. On reception, all samples were stored at –80 °C while awaiting processing. All RT-qPCR tests were performed according to directions from the laboratory of the University of Liège, which elaborated the collecting device and technique, as described in Saegerman et al. [21]. RT-qPCR results are reported as values of cycle threshold (Ct value, i.e., as defined by Public Health England, a semi-quantitative indicator of the concentration of viral genetic material in a sample [23]). In the study, positive samples were arbitrarily classified into three categories of Ct Values as follows: <25, 25–30, >30. 

Phylogenetic analysis was performed on positive saliva samples presenting Ct values < 25 to investigate transmission of virus between participants. Total nucleic acid was extracted using the MagMAX™ Viral/Pathogen II Nucleic Acid Isolation Kit according to the manufacturer instructions (Cat. No. A48383, ThermoFisher Scientific, Waltham, MA, USA). The amplicon-based Illumina COVIDSeq protocol (Illumina Inc., San Diego, CA 92122, USA) in combination with the ARTIC v4 primers pools (https://artic.network/, accessed on 25 August 2022) was used for sequencing according to manufacturer’s instructions. The pooled library was diluted to a final concentration of 100pM for a single read (1 × 150 bp) sequencing on a NextSeq 1000 instrument. Generated fastq files were uploaded on the cloud-based ASP-IDNS^®^−5 analysis software (SmartGene, 1015 Lausanne, Switzerland). For analysis we used the “SARS-CoV-2 full genome” pipeline version 2.5.0_COV_v0.2. Online Nextclade version 2.3.0 software as a first sequence aligner, allowing comparison to the Wuhan-hu-1/2019 (MN908947) SARS-CoV-2 reference genome and permitting a clade assignment (https://clades.nextstrain.org, accessed on 25 August 2022). FASTA sequences were also submitted to the Pangolin (4.1.1) COVID-19 Lineage Assigner. Phylognentic Tree was generated by submitting the Fasta files to the NGPhylogeny web interface [24]. The workflow included: sequence alignment using the MAFFT software, curation of the sequences with the block mapping and gathering with entropy (BMGE) software, tree generation using the fast distance-based phylogeny inference program FastME 2.0, and tree output formatted with the Newick display. A detailed description of SARS-CoV-2 whole-genome sequences method is available in the Appendix A.

All participants as well as study staff were blinded to RT-qPCR results, until the end of the study, as per protocol. Therefore, participation in the study and PCR test results had no impact on school attendance. During the study period, school attendance was, however, subject to health measures dictated by the Belgian government for all participants. These mitigation measures changed over time. At the time of the study, sanitary measures in elementary schools were as follows: frequent hand hygiene, mask wearing for adult staff when in close contact with pupils or other staff, quarantine measures for symptomatic individuals and high-risk contacts until PCR results from samples taken by the individual’s healthcare provider, and finally, closure of classes if two pupils or the teacher had confirmed SARS-CoV-2 cases. Therefore, no symptomatic individuals should have been tested in this study.

In addition, we cross-referenced positive PCR results with the cases reported to the school health promotion department (SHPD) in the participating schools. During the pandemic, the SHPD was designated to carry out surveillance of COVID-19 cases in schools and perform contact tracing in agreement with the regional public health units. SHPD reported the weekly numbers of pupils and staff with confirmed SARS-CoV-2 infections and collected data on the suspected source of infection (index case inside or outside the school) and counted probable secondary cases, i.e., having been infected in the school environment. 

### 2.3. Statistical Analysis

Because this was a pilot study, no formal sample size determination was performed. Categorical variables are described by counts and percentages and continuous variables by means and standard deviations or median and interquartile ranges (IQR) for non-normal distributions. Data were analyzed using STATA software (version 14.1 StataCorp LP, College Station, TX, USA). 

The protocol was approved by the Ethical Committee of the Cliniques universitaires Saint-Luc, UCLouvain (approval number 2020/16NOV/552, approved on the 20/11/20) and was registered on clinicaltrial.gov (Number NCT05046470) and on ISRCTN (Number ISRCTN16837012).

## 3. Results

We conducted this study between 14 January 2021 and 18 May 2021.

All schools were selected according to the above-mentioned criteria. Among eight schools that were invited to participate, two schools were excluded due to a high rate of refusal from either the staff or both the pupils and staff. Subsequently, five new schools were invited, resulting in a total of 11 included schools out of 13. The geographical distribution of the 11 schools included is depicted in Figure 2.

In total, 932 children and 242 school staff were included and completed the study, which corresponds to an overall participation rate of 37.5% and 54.7%, respectively (Figure 3, Table 1). Participation varied between schools, ranging from 10.4% to 71.1% and from 20.0% to 100% in children and school staff, respectively (Table 1). 

The different schools were included sequentially over time. Six schools participated from January week 2 2021 to March week 10 2021, including one week of school break. Four schools started in February on week 8 of 2021 and ended in April on week 17 of 2021, with a 4-week period of school break, and finally one school participated from week 11 March 2021 to May week 20 2021 (Figure 4). 

Among 1162 available serological tests (6 missing and 6 invalid) performed once on study inclusion, 191 children and 61 staff tested seropositive, which corresponds to a positivity rate of 20.7% (95% CI 18.2–23.4) and 25.4% (95% CI 20.5–31.5), respectively, in children and school staff (Table 1) for whom a valid result is available. 

Over the whole study period which included six weekly visits to each of the schools, a total of 6449 saliva samples were obtained; 5226 from children and 1217 from school staff, which corresponds to an average of 5.6 tests per child and 5.0 tests per adult. Some samples are missing due to absenteeism of participants on the collection date or due to invalid samples (swallowing or wrong execution) (Table 2). During the entire study period, 44 (0.7%) SARS-CoV-2 PCR tests were positive, 29 in children and 6 in school staff, which corresponds to a positivity rate of 3.11% in children and 2.46% in school staff. Of these children, nine had a positive test for two consecutive weeks (Table 3). Among all PCR-positive tested participants, 27/29 children (93.1%) and 2/6 (33.3%) school staff had a negative serological test at the beginning of the study. In four schools, we did not detect any positive PCR tests, and in 8 out of 11 schools, there were no positive PCR tests in school staff. 

In school n°8, four cases were detected concomitantly in the same class (first grade) and had a positive PCR SARS-CoV-2 for 2 weeks during visits 3 and 4. In the same school, three other positive cases were detected in the same class (4th grade), on visit 2. No school staff member participating in the study tested positive in this school during the study period (Table 3). In school n°7, two cases were also detected in the same 4th grade class. All the other positive cases were isolated. 

Cross-referencing the data from our study with information from the SHPD, there was no detectable increase in cases during the study period or in a 2-week time span after the end of the study period in all schools except in school n°2 (Figure 5). In school n°2, the SHPD declared a total of 36 cases between week 9 and week 18 of year 2021; 32 pupils and 4 school staff members (Figure 5, Appendix A). Among the 32 pupils detected, 14 who were considered as index cases for secondary cases within the school were themselves infected outside the school area, and 18 were considered as secondary cases, i.e., having been infected themselves in the school environment. The four school staff members who tested positive are considered as index cases. Due to the large number of cases detected in the school, first-grade classes were closed during weeks 11 and 12, the second grade on week 12, as well as one class of the third grade. 

Next-generation-sequencing analyses of the positive PCR results could be performed in 10 positive PCR with a Ct of less than 25. Unfortunately, only 57% of the genome analyzed was common to all 10 samples, allowing phylogenetic analyses to be performed on a very fragmented genome. The results obtained under these conditions suggested that the ten strains differ, as shown on Figure 6. Only two viruses seemed close, but participants came from different schools. 

## 4. Discussion

A better understanding of SARS-CoV-2 transmission in schools is of the utmost importance to guide future decisionmakers on school closures and educational disruptions, actions that have implications far beyond physical health. As mentioned above, the role of children in the dynamics of transmission SARS-COV-2 during the different stages of the pandemic and under the different variants of concern remains controversial. Since the circulation of more transmissible variants, it seems clear that children and young adolescents are as prone as older individuals to infection but less likely to develop a symptomatic or severe infection [1,2,3,25]. The question of whether these young populations are naturally less infectious than older age groups is still debated [25]. Moreover, as they experience less symptomatic disease, they can more easily escape testing strategies and are considered as potential silent drivers of the infection [26]. The singular design of this study, being weekly sampling during consecutive weeks with no exclusion of positive but asymptomatic pupils and adult staff is suited to estimate the potential and effective role of asymptomatic cases in transmission within the classes and the schools of elementary education.

Overall, the literature shows that the risk of transmission to and from children in schools seems to be low, especially in elementary education [27,28,29]. Our observational uncontrolled prospective study supports a low secondary transmission in schools. Among 11 selected schools, only 44 (0.7%) repetitively performed SARS-CoV-2 RT-qPCR tests came back positive, in 29 children and 6 school staff. These data are in line with similar observations made in schools. Vogel and colleagues, in Germany, reported that 8 out of 23,905 samples were positive in 17 elementary schools despite high community incidence rates [30]. In the UK, Ladhani and colleagues concluded that SARS-CoV-2 infection rates were low in primary schools following partial and full reopening of schools [31].

Our study detected silent cases, being asymptomatic cases that escaped the detection system in place in Belgium at that time. Indeed, they were not picked up through the School Health Promotion Department (SHPD). Among our positive cases, 20/29 pupils were isolated cases in a class and 2 were isolated cases in the school. There was no increase in the number of cases detected in the following weeks, either. One can argue that not all individuals attending the different schools were participating in the study. However, even if we could have missed secondary cases, data do not support silent transmission in school resulting in apparent ill cases, as no increase in cases was detected by SHPD, up to a 2-week period after the end of the study. Furthermore, we did not observe many clusters, as only one school (school n°8) had two classes with, respectively, four and three pupils who tested positive on the same visit. Inside these classes, the classmates participating in the study did not become infected on the following visits. When we cross-referenced our data with the SHPD, we found that their data are in line with our observation. No school staff from this school had a positive test during the study period. This could have contributed to low secondary transmission, as transmission is more likely to occur if the index case is a teacher rather than a child, other factors being equal [29]. According to these results, we estimate that over-testing of asymptomatic children who have been in contact with confirmed SARS-CoV-2-positive individuals is probably not necessary. Indeed, most cases detected in our study were isolated cases (except for school 8). 

The size of the study was not formally calculated but it had enough power to detect a significant transmission rate. Using a purposive sampling, the aim of the study was to find an index case in a class and then at least one new case in the same class at the next visits. With 932 children across 119 classes, the mean number of children per school class was 8. We observed at least one case in 18 school classes before visit 6, and there was no new case in the same class in subsequent visits in 17 out of these 18 classes. Only school class 4B in School 7 had a case at visit 2 followed by a new case at visit 5, one month later (Table 3 and Figure 4). To observe 0 new cases in 7 classmates, 17 times out of 18, and 1 new case in 7 classmates 1 time out of 18 leads to a likelihood over 95% that the transmission probability is lower than 0.009. The size of our study was large enough to detect such a low transmission rate.

Unfortunately, due to low viral loads (low Ct values) and due to technical issues, phylogenetic analyses could be performed only in 10 positive saliva samples and concerned just 57% of the genome, a percentage common to all 10 samples. The results obtained in these limiting conditions suggested that individuals were infected outside the school, but we do not have the power to confirm that. 

Additionally, schools seem to reflect the evolution of transmission in society [32]. During our study period, the dominant strain was the variant of concern B.1.1.7 (Alpha), and SARS-CoV-2 circulation in Belgium was low. The positivity rate in our study was 0.7%. In one school, we observed two clusters, in March 2021. These results might simply reflect the multitude of factors involved in the transmission, including the intertwined relationship between community cases and school-acquired cases. Indeed, data from Sciensano, assessing the weekly incidence of SARS-CoV-2 in Belgium at that time, showed an ongoing increase in the number of positive SARS-CoV-2 in the whole country during that period, starting on February the 26th. 

In our study protocol, a salivary test was preferred over nasopharyngeal swabbing. This non-invasive method seems better accepted for repeated testing in children, as highlighted by Vogel and colleagues [30]. On site, salivary tests for weekly surveillance were well accepted by participants. Sampling was performed in the setting of a scientific study and, therefore, was supervised by study staff. As the weeks went by, children were used to the method and only a few tests were lost due to swallowing or spitting wrongly outside the container. We propose that this less invasive method should be the norm when testing is performed in the pediatric population. 

Our study has limitations. First, the participation rate in schools varied widely between schools and was never exhaustive. Recruitment is always a challenge; however, in the time of the pandemic, sanitary measures further complicated recruitment. Access to information was not uniform in the population, and the virtual communication with the targeted population impeded participation. Second, the phylogenetic analysis was inconclusive due to the limited number of analyzed samples and the limitations of the technique as well as of the obtained results. The lack of this information prevents us from asserting the circulation of the same virus within a class or a school. Third, this study took place during a low incidence period of COVID-19 cases in Belgium, and therefore, of low circulation of the virus. In January 2021, at the beginning of our study, the incidence rate for the last 14 days, in Belgium, was 205 for 100,000 inhabitants, with a positivity rate of 6.7%. In May 2021, at the end of our study period, the incidence was 355 for 100,000 inhabitants, with a positivity rate of 6.3%. In May 2021, the immunization rate for all individuals of more than 18 years of age was 12.4%, and, for the population of more than 65 years old, this was 27.7%. The dynamics of COVID-19 transmission changed quickly after the end of this study, with the appearance of the Delta variant inducing a wave from late July 2021 until early December 2021, followed by the Omicron wave [33]. As consequence, due to the higher transmissibility of those variants, the incidence of COVID-19 in Belgium also increased significantly [19]. During the wave of Fall 2020, the COVID-19 incidence in the school population was considerably lower than in the general population [18,33]. This figure has changed during the wave of Fall 2021 with a recorded incidence in children aged 0–9 years old 5.3 times higher than during the same period 1 year before [33]. 

## 5. Conclusions

Globally, in elementary schools, the positivity rate of repetitive weekly PCR sampling for SARS-CoV-2 in asymptomatic individuals was low during our study period. While asymptomatic positive cases were detected, children remaining present in class during the Alpha variant wave of the pandemic did not lead to an increased number of secondary cases or clusters in their class or school. These data strengthen the view that keeping elementary education open at that point was a balanced decision, especially when we counterbalance the detrimental effect of disrupting education and the negative impact on the well-being of children of school closure. However, it would be interesting to conduct the same study during a period of high incidence, or during the circulation of other VOCs.

## Figures and Tables

**Figure 1 viruses-14-02199-f001:**
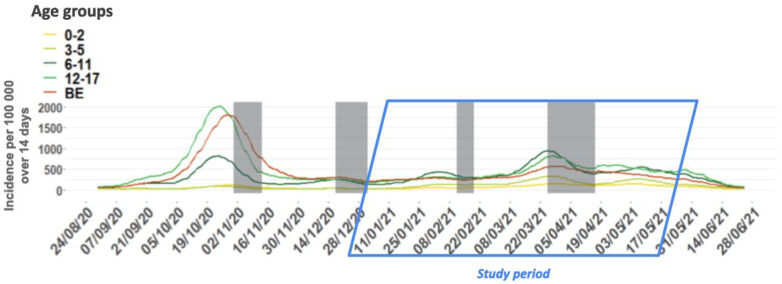
Fourteen-day incidence of COVID-19 cases per 100,000 persons by age group and for the general population, from 1 September 2020 to 30 June 2021, Belgium. The incidence is represented by the sum of the number of cases in the previous 14 days over the number of people in that age group. “BE” indicates the whole population in Belgium (all age groups). Shaded areas indicate school holiday periods.

**Figure 2 viruses-14-02199-f002:**
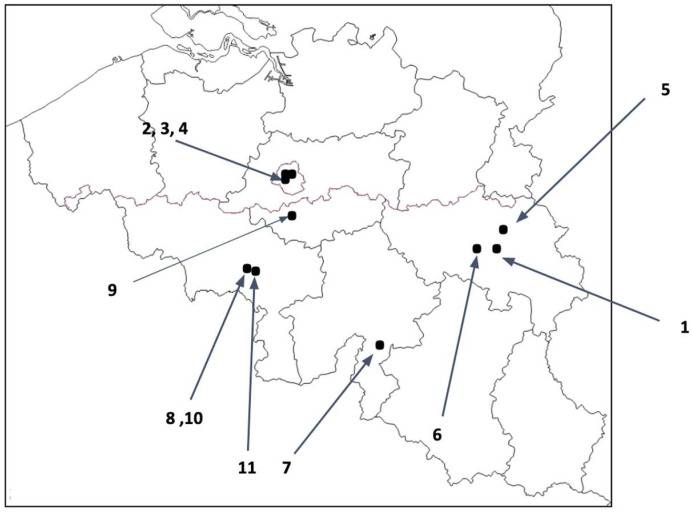
Geographical distribution of primary schools included in the study.

**Figure 3 viruses-14-02199-f003:**
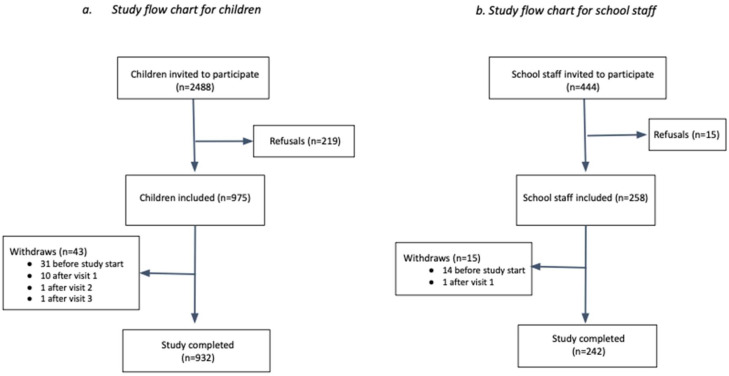
Study flowchart.

**Figure 4 viruses-14-02199-f004:**
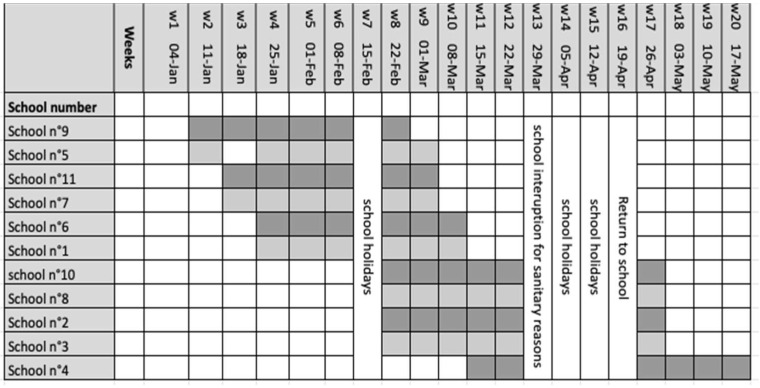
Study timeline. Sequentially inclusion of schools in the study. The weeks mentioned correspond to the calendar weeks of the year 2021. Shaded areas indicate the weeks of school visits. School n°5 was closed (optional school holiday) on the day of the study staff’s visit in week 3.

**Figure 5 viruses-14-02199-f005:**
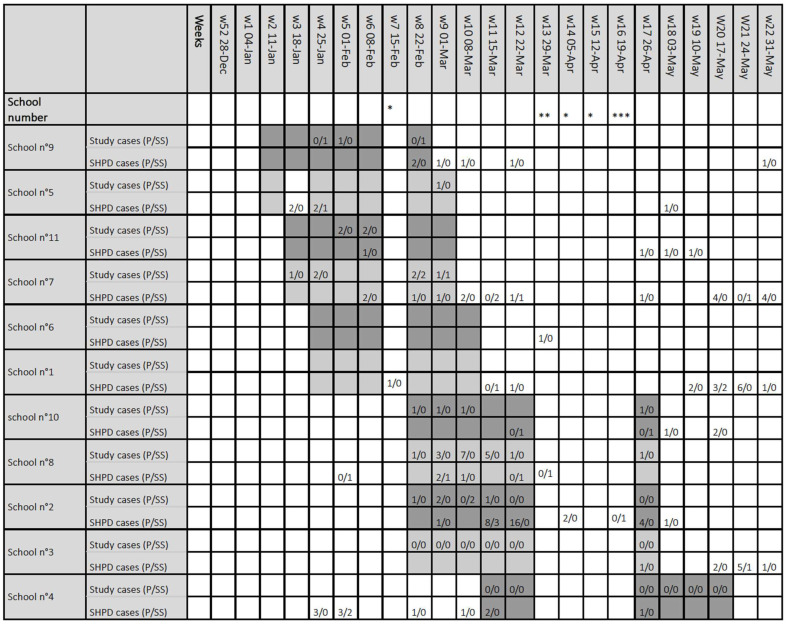
Reported cases of positive SARS-CoV-2 PCR during the study period detected by the study samples and reported by health promotion department from each school. SARS-CoV-2 cases detected by the study DynaTracs (first line) and reported by the school health promotion department (second line). Distinction was made between pupils (p) and adults school staff (SS). The weeks mentioned correspond to the calendar weeks of the year 2021. Shaded areas indicate the weeks of school visits. * School holidays, ** School Interruption for sanitary reasons, *** Return to school. p: pupils, SS: school staff, SHPD: school health promotion department.

**Figure 6 viruses-14-02199-f006:**
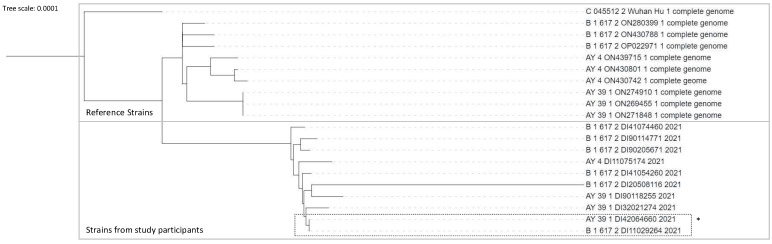
Phylogenetic tree comparing reference strains including Wuhan strain and strains from 10 participants. The upper part of the diagram represents the results from reference strains. The shaded part of the diagram represents results from the study samples. * Two strains with similarities, - participants coming from different schools.

**Table 1 viruses-14-02199-t001:** Description of schools, participation rate, seropositivity rate at inclusion and number of positive SARS-CoV-2 PCR tests.

School Number	Cumulative Incidence of SARS-CoV-2	School Size	SES	Children Participation Rate *n* (%)	School StaffParticipation Rate *n* (%)	Cases with a Positive SARS-CoV-2 PCR (*n*)	Seropositivity Rate at Study Onset *n* (%)
1	High	Large	High	125/244 (51.2)	29/42 (69.0)	0/125 children 0/29 Staff	Children 24/125 (19.2) Staff 12/29 (41.4)
2			Low	44/379 (11.6)	17/47 (36.2)	4/45 children 2/17 Staff	children 5/44 (11.1) Staff 2/17 (11.8)
3				78/256 (30.5)	32/35 (91.4)	0/78 children 0/32 Staff	children 19/78 (24.4) Staff 6/32 (18.8)
4				26/249 (10.4)	17/45 (37.8)	0/26 children 0/17 Staff	children 6/24 (25.0) Staff 8/17 (47.1)
5		Small	High	94/165 (57.0)	16/42 (38.1)	1/94 child 0/16 Staff	children 20/93 (21.5) Staff 8/16 (50.0)
6			Low	41/125 (32.8)	27/38 (71.1)	0/41 children 0/27 Staff	children 6/41 (14.6) Staff 4/25 (16.0)
7	Low	Large	High	151/298 (50.7)	32/45 (71.1)	6/151 children 2/32 Staff	children 30/148 (20.3) Staff 11/32 (34.4)
8			Low	99/275 (36.0)	10/50 (20.0)	12/99 children 0/10 Staff	children 22/98 (22.5) Staff 0/10 (0.0)
9		Small	High	150/211 (71.1)	28/39 (71.8)	1/150 child 2/29 Staff	children 29/148 (19.6) Staff 5/28 (17.9)
10			Low	97/227 (42.7)	23/50 (46.0)	3 /97 children 0/23 Staff	Children 19/96 (19.8) Staff 1/23 (4.3)
11				27/59 (45.9)	11/11 (100)	2/27 children 0/11 Staff	children 11/27 (40.7) Staff 4/11 (36.4)

Description of participating schools according to 3 criteria: (i) school size (small < 230 pupils), (ii) socio-economic status (SES) of the population attending the school (high level score ≥ 13), and (iii) cumulative incidence of SARS-CoV-2 infection in the geographic province of the school during the first wave of pandemic, in spring 2020 (low incidence < 5/1000 persons). Participation of children and school staff are represented by total number and percentages. Positive SARS-CoV-2 PCR are represented by total numbers with the distinction between children and school staff (SS). Positive serological tests of children and school staff are represented by total number and percentages. Missing data or invalid tests are excluded from the analyses. SES: socio-economic status; Staff: school staff.

**Table 2 viruses-14-02199-t002:** Summary table of SARS-CoV-2 PCR results for each school.

School Number/Participants (*n*)		Visit 1			Visit 2			Visit 3			Visit 4			Visit 5			Visit 6		
		PCR−	PCR+	M	PCR−	PCR+	M	PCR−	PCR+	M	PCR−	PCR+	M	PCR−	PCR+	M	PCR−	PCR+	M
1/154	Week n°	w4			w5			w6			w8			w9			w10		
	Pupils SS Total	116 29 145	0 0 0	1 0 1	119 29 148	0 0 0	4 0 4	117 29 146	0 0 0	6 0 6	123 28 151	0 0 0	2 1 3	119 25 144	0 0 0	6 4 10	119 28 147	0 0 0	6 1 7
	Total	146			152			152			154			154			154		
2/62	Week n°	w8			w9			w10			w11			w12			w17		
	Pupils SS Total	42 11 53	1 0 1	0 0 0	41 17 58	2 0 2	2 0 2	43 13 56	0 2 2	2 2 4	26 9 35	1 0 1	18 8 26	30 5 35	0 0 0	15 12 27	36 10 46	0 0 0	9 7 16
	Total	54			62			62			62			62			62		
3/110	Week n°	w8			w9			w10			w11			w12			w17		
	Pupils SS Total	77 31 108	0 0 0	0 0 0	71 28 99	0 0 0	7 4 11	77 26 103	0 0 0	1 6 7	74 23 97	0 0 0	4 9 13	69 16 85	0 0 0	9 16 25	75 17 92	0 0 0	3 15 18
	Total	108			110			110			110			110			110		
4/43	Week n°	w11			w12			w17			w18			w19			w20		
	Pupils SS Total	22 15 37	0 0 0	0 0 0	24 14 38	0 0 0	0 3 0	20 4 24	0 0 0	6 13 19	26 12 38	0 0 0	0 5 5	24 8 32	0 0 0	2 9 11	24 8 32	0 0 0	2 9 11
	Total	37			41			43			43			43			43		
5 /110	Week n°	w2			w4			w5			w6			w8			w9		
	Pupils SS Total	90 16 106	0 0 0	0 4 4	67 12 79	0 0 0	27 4 31	87 13 100	0 0 0	7 3 10	89 13 102	0 0 0	5 3 8	94 15 109	0 0 0	0 1 1	92 13 105	1 0 1	1 3 4
	Total	110			110			110			110			110			110		
6/68	Week n°	w4			w5			w6			w8			w9			w10		
	Pupils SS Total	41 24 63	0 0 0	0 2 2	39 24 63	0 0 0	2 3 5	39 24 63	0 0 0	2 3 5	40 25 65	0 0 0	1 2 3	39 24 63	0 0 0	2 3 5	40 25 65	0 0 0	1 2 3
	Total	65			68			68			68			68			68		
7/183	Week n°	w3			w4			w5			w6			w8			w9		
	Pupils SS Total	148 32 180	1 0 1	1 0 1	144 30 174	2 0 2	5 2 7	150 32 182	0 0 0	1 0 1	136 29 165	0 0 0	15 3 18	140 25 165	2 2 4	9 5 14	138 29 167	1 0 1	12 3 15
	Total	182			183			183			183			183			183		
8/109	Week n°	w8			w9			w10			w11			w12			w17		
	Pupils SS Total	91 10 101	1 0 1	1 0 1	91 9 100	3 0 3	4 1 5	88 10 98	7 0 7	4 0 4	81 9 90	5 0 5	13 1 14	84 8 92	1 0 11	14 2 16	88 7 95	1 0 1	10 3 13
	Total	103			108			109			109			109			109		
9/179	Week n°	w2			w3			w4			w5			w6			w8		
	Pupils SS Total	149 26 175	0 0 0	1 1 2	145 21 166	0 0 0	5 7 12	142 21 163	0 1 1	8 7 15	144 24 168	1 0 1	5 5 10	143 23 166	0 0 0	7 6 13	143 19 162	0 1 1	7 9 16
	Total	177			178			179			179			179			179		
10/120	Week n°	w8			w9			w10			w11			w12			w17		
	Pupils SS Total	91 23 114	1 0 1	0 0 0	89 20 109	1 0 1	6 3 9	88 23 111	1 0 1	7 0 7	92 22 114	0 0 0	5 1 6	89 21 110	0 0 0	8 2 10	84 22 106	1 0 1	12 1 13
	Total	115			119			119			120			120			120		
11/38	Week n°	w3			w4			w5			w6			w8			w9		
	Pupils SS Total	26 9 35	0 0 0	0 1 1	27 10 37	0 0 0	0 1 1	24 10 34	2 0 2	1 1 2	14 8 22	1 1 2	11 3 14	23 6 29	0 0 0	4 5 9	26 9 35	0 0 0	1 2 3
	Total	36			38			38			38			38			38		

N: total number; M: missing or invalid sampling; SS: school staff. The weeks mentioned correspond to the calendar weeks of the year 2021.

**Table 3 viruses-14-02199-t003:** Description of positive SARS-CoV-2 PCR cases in the study population by week of the study.

Children									
Case Number	School Number	Serology Result	School Level	SARS-CoV-2 PCR					
				Visit 1	Visit 2	Visit 3	Visit 4	Visit 5	visit 6
1	school 9	neg	3C	-	-	-	C_t_ < 25	-	-
2	school 5	neg	4A	-	-	-	-	-	C_t_ ≥ 30
3	school 11	neg	1A	-	-	C_t_ ≥ 30	C_t_ ≥ 30	-	-
4	school 11	neg	5A	-	-	C_t_ 25–30	C_t_ ≥ 30	-	-
5	school 7	neg	1A	-	-	-	-	-	C_t_ ≥ 30
6	school 7	neg	2B	-	C_t_ < 25	-	-	-	-
7	school 7	neg	3B	-	-	-	-	C_t_ 25-30	-
8	school 7	neg	4B	-	C_t_ < 25	-	-	-	-
9	school 7	pos	4B	-	-	-	-	C_t_ ≥30	-
10	school 7	pos	6B	C_t_ ≥ 30	-	-	-	-	-
11	school 10	neg	1B	C_t_ < 25	C_t_ ≥ 30	-	-	-	-
12	school 10	neg	4B	-	-	-	-	-	C_t_ ≥ 30
13	school 10	neg	6A	-	-	C_t_ 25–30	-	-	-
14	school 8	neg	1A	-	-	C_t_ 25–30	C_t_ ≥ 30	-	-
15	school 8	neg	1A	-	-	C_t_ 25–30	C_t_ ≥ 30	-	-
16	school 8	neg	1A	-	-	C_t_ 25–30	missing	-	-
17	school 8	neg	1A	missing *	-	C_t_ ≥ 30	C_t_ < 25	missing	-
18	school 8	neg	3A	C_t_ ≥ 30	-	-	-	-	-
19	school 8	neg	3B	-	-	-	C_t_ < 25	-	-
20	school 8	neg	4A	-	C_t_ < 25	C_t_ 25–30		-	-
21	school 8	neg	4A	-	C_t_ < 25	missing	Invalid test	C_t_ 25–30	-
22	school 8	neg	4A	-	C_t_ < 25	C_t_ 25–30	-	-	-
23	school 8	neg	4B	-	-	C_t_ ≥ 30	-	-	-
24	school 8	neg	6A	-	-	-	C_t_ 25–30	missing	-
25	school 8	neg	6B	-	-	-	-	-	Ct < 25
26	school 2	neg	1A	-	C_t_ < 25	-	missing	missing	-
27	school 2	neg	2C	-	C_t_ ≥ 30	-	missing	-	-
28	school 2	neg	4C	-	-	-	C_t_ 25–30	-	-
29	school 2	neg	5A	C_t_ 25–30	-	-	-	-	missing
**School Staff**									
1	school 9	neg	NA	-	-	-	-	-	C_t_ 25–30
2	school 9	pos	NA	-	-	C_t_ ≥ 30	-	-	-
3	school 7	neg	NA	-	-	-	-	C_t_ 25–30	-
4	school 7	pos	NA	-	-	-	-	C_t_ 25–30	-
5	school 2	neg	NA	missing *	-	C_t_ < 25	missing	missing	-
6	school 2	neg	NA	missing *	-	C_t_ < 25	missing	missing	missing

Numbers were assigned chronologically according to the dates of sample collection, separating positive samples from pupils and school staff. Results are categorized into 3 categories of C_t_ value: C_t_ < 25, C_t_: 25–30, C_t_ >30. Missing data correspond either to the fact that the participant was absent on the day of the visit or that the sample could not be taken for technical reasons (swallowed sample or inability to spit). * Inclusion at visit 2, NA: not applicable, C_t_: cycle threshold.

## Data Availability

J.F., O.C., K.C., D.V.d.L. and A.R. had full access to all data and took responsibility for the integrity of the data.

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
