# Peer review of "SARS-CoV-2 Transmission in Belgian French-Speaking Primary Schools: An Epidemiological Pilot Study"

_viruses, 2022, doi:10.3390/v14102199_

Round 1

Reviewer 1 Report

Review of the paper "SARS-CoV-2 transmission in Belgian French-speaking primary 2 schools. An epidemiological pilot study".

In this paper, the authors describe a prospective study conducted in some schools in Belgium in order to evaluate the transmission of COVID in primary school pupils and teachers at a time of low incidence of the disease. The aim of this study is to support the idea that closing schools has no significant effect on the incidence of the disease.

The study and procedures are well described, and the webpage https://www.sesa.ucl.ac.be/Dynatracs/ has been helpful in understanding the details of the study. Furthermore, the study provides information about the clades circulating and supports the utility of certain techniques to obtain samples to test COVID in children.

The only concern about the study, and it is mentioned by the authors, is that no formal sample size determination was performed. Maybe it may be done a posteriori to check if the required sample size was fulfilled.

Other minor issues are:

* On page 8, the paragraph "Among 1162 available serological tests ..." is a bit confusing. If the study includes 932 children and 242 school staff, does it mean that only one test has been done on each individual? This is contradictory to the next paragraph where " ... a total of 6449 saliva samples were obtained ...". Also, the positivity rates seem to be referred to the number of people instead of the number of tests. Please, clarify.

* In Table 3, Ct appears as the cycle threshold, and it does not seem to be defined.    

Reviewer 2 Report

the work is interesting and well illustrated, except that salivary testings currently out of place for the diagnosis of infection from SARSCOV2

The work describes the likelihood of contagion from Covid 19 infection in some schools in Belgium and France during the pandemic period

The swab used is the salivary one to highlight positive covid cases and any infections, for which it was practiced weekly between students and school staff.

The salivary swab has not come into use in the diagnosis of covid 19 infection but is well tolerated

The study shows that the probability of getting infected is low in schools but it would have been interesting to be able to carry out the nasal pharyngeal swab to verify the real cases.

the strong point is that the numbers are high and the observation is long.

The weak point is the lack of comparison between the salivary swab and the nasopharyngeal swab method for the diagnosis of covid 19, being, as mentioned before, salivary tests are no longer practiced for the diagnosis of covid-19 infection.
